# A Good Representation Detects Noisy Labels

## Abstract

Label noise is pervasive in real-world datasets, which encodes wrong correlation patterns and impairs the generalization of deep neural networks (DNNs). It is critical to find efficient ways to detect the corrupted patterns. Current methods primarily focus on designing robust training techniques to prevent DNNs from memorizing corrupted patterns. This approach has two outstanding caveats: 1) applying this approach to each individual dataset would often require customized training processes; 2) as long as the model is trained with noisy supervisions, overfitting to corrupted patterns is often hard to avoid, leading to performance drop in detection. In this paper, given good representations, we propose a universally applicable and training-free solution to detect noisy labels. Intuitively, good representations help define "neighbors" of each training instance, and closer instances are more likely to share the same clean label. Based on the neighborhood information, we propose two methods: the first one uses "local voting" via checking the noisy label consensuses of nearby representations. The second one is a ranking-based approach that scores each instance and filters out a guaranteed number of instances that are likely to be corrupted, again using only representations. Given good (but possibly imperfect) representations that are commonly available in practice, we theoretically analyze how they affect the local voting and provide guidelines for tuning neighborhood size. We also prove the worst-case error bound for the ranking-based method. Experiments with both synthetic and real-world label noise demonstrate our training-free solutions are consistently and significantly improving over most of the training-based baselines.

## 1 Introduction

The generalization of deep neural networks (DNNs) depends on the quality and the quantity of the data. Nonetheless, in practice real-world datasets often contain label noise that challenges the above assumption (Krizhevsky et al., 2012; Zhang et al., 2017; Agarwal et al., 2016). Employing human workers to cleaning annotations is one reliable way to improve the label quality, but it is too expensive and time-consuming for a large-scale dataset. One promising way to automatically cleaning up label errors is to first algorithmically detect possible label errors from a large-scale dataset (Cheng et al., 2021; Northcutt et al., 2021a; Pruthi et al., 2020; Bahri et al., 2020), and then correct them using either algorithm or crowdsourcing (Northcutt et al., 2021b).

Almost all the algorithmic detection approaches focus on designing customized training processes to learn with noisy labels, where the idea is to train DNNs with noisy supervisions and then make decisions based on the output (Northcutt et al., 2021a) or gradients (Pruthi et al., 2020) of the last logit layer of the trained model. The high-level intuition of these methods is the memorization effects (Han et al., 2020), i.e., instances with label errors, a.k.a., corrupted instances, tend to be harder to be learned by DNNs than clean instances (Xia et al., 2021; Liu et al., 2020). By setting appropriate hyperparameters to utilize the memorization effect, corrupted instances could be identified.

The above methods suffer from two major limitations: 1) the customized training processes are task-specific and may require fine-tuning hyperparameters for different datasets/noise; 2) as long as the model is trained with noisy supervisions, the memorization of corrupted instances exists. The model will "subjectively" and wrongly treat the memorized/overfitted corrupted instances as clean. For example, some low-frequency/rare clean instances may be harder to memorize than high-frequency/common corrupted instances. Memorizing these corrupted instances lead to unexpected and disparate impacts (Liu, 2021). One way to avoid memorizing/overfitting is to drop the depen-

dency on training using the noisy supervisions, which motivates us to design a *training-free* method to find label errors. Intuitively, we can carefully use the information from nearby representations to infer whether one instance is corrupted or not. This observation inspired our solution of using a good representation to first identify "neighbor" instances for each training example, whose noisy labels jointly would help us perform the detection. Note the representation extractor could be fully independent of noisy supervisions by referring to unsupervised learning or self-supervised learning, or even adapting from other tasks.

Our training-free method enables more possibilities beyond a better detection result. For example, the concerns of the required assumptions and hyperparameter tuning in those training-based methods will now be released due to our training-free property. The complexity of such a solution will also be much lower, again due to the removal of the possibly involved training processes. This light detection solution also has the potential to serve as a pre-processing module to prepare data for other sophisticated tasks (e.g., semi-supervised learning (Xie et al., 2019; Berthelot et al., 2019)).

Our main contributions are: 1) *New perspective:* Different from current methods that train customized models on noisy datasets, we proposed a training-free solution to efficiently detect noisy labels given good representations. We believe this is the first attempt of the same kind to the best of our knowledge. 2) *Efficient algorithms:* Based on the neighborhood information, we propose two methods: a voting-based local detection method that only requires checking the noisy label consensuses of nearby representations, and a ranking-based global detection method that scores each instance by its likelihood of being clean and filters out a guaranteed percentage of instances with low scores as corrupted ones. 3) *Theoretical analyses:* We theoretically analyze how a good representation (but possibly imperfect in practice) affects the local voting and provide guidelines for tuning neighborhood size. We also prove the worst-case error bound for the ranking-based method. 4) *Numerical findings:* Our numerical experiments show three important messages: in label noise detection, i) training with noisy supervisions may not be necessary; ii) representation layers tend to be more useful than the logit layers; iii) representations from other tasks or domains are helpful.

## 1.1 RELATED WORKS

**Learning with noisy labels** There are many other works that can detect corrupted instances (a.k.a. sample selection) in the literature, e.g., (Jiang et al., 2018; Han et al., 2018; Yu et al., 2019; Yao et al., 2020; Wei et al., 2020; Jiang et al., 2020; Zhang et al., 2021), and its combination with semi-supervised learning (Nguyen et al., 2019; Li et al., 2020b; Cheng et al., 2021). Another line of works focus on designing robust loss functions to mitigate the effect of label noise, such as numerical methods (Ghosh et al., 2017; Zhang & Sabuncu, 2018; Gong et al., 2018; Amid et al., 2019; Wang et al., 2019; Shu et al., 2020) and statistical methods (Natarajan et al., 2013; Liu & Tao, 2015; Patrini et al., 2017; Liu & Guo, 2020; Xia et al., 2019; 2021; Wei & Liu, 2021; Zhu et al., 2021a; Li et al., 2021; Liu et al., 2020). All these methods require training DNNs with noisy supervisions.

**Pre-training or Self-supervised Learning** Training DNNs from scratch may be time-consuming (Goyal et al., 2017). A common strategy to alleviate this issue is initializing DNN with a pre-trained model when training on new tasks (Krizhevsky et al., 2012; Zeiler & Fergus, 2014; Sermanet et al., 2013). The pre-trained model can be used for down-stream tasks due to the great transfer-ability of DNNs (Tan et al., 2018). Popular pre-trained models include BERT (Devlin et al., 2018) for language tasks or CLIP (Radford et al., 2021) for vision-language tasks. Recently, unsupervised learning (Ji et al., 2019) and self-supervised learning (Jaiswal et al., 2021; Liu et al., 2021; He et al., 2020; Chen et al., 2020) also exhibit great power when fine-tuning DNN on down-stream tasks.

**Label aggregation** Our work is also relevant to the literature of crowdsourcing that focuses on label aggregation (to clean the labels)(Liu et al., 2012; Karger et al., 2011; 2013; Liu & Liu, 2015; Zhang et al., 2014). Most of these works can access multiple reports (labels) for the same input feature, while our real-world datasets usually have only one noisy label for each feature.

## 2 PRELIMINARIES

**Instances** Traditional classification problems with perfect supervisions build on a clean dataset $D := \{(x_n, y_n)\}_{n \in [N]}$, where $[N] := \{1, 2, \cdots, N\}$. Each *clean instance* $(x_n, y_n)$ includes *feature* $x_n$ and *clean label* $y_n$, which is drawn according to random variables $(X, Y) \sim \mathcal{D}$. In a more practical case, the clean labels are likely to be unavailable and the learner could only observe a noisy

Figure 1: Detect noisy labels with similar representations. Orange circle: instance with noisy label 1. Blue square: instance with noisy label 2. Green dashed circle: A $k$-NN example.

dataset denoted by $\widetilde{D} := \{(x_n, \tilde{y}_n)\}_{n\in[N]}$, where $(x_n, \tilde{y}_n)$ is a *noisy instance* and the *noisy label* $\tilde{y}_n$ may or may not be identical to $y_n$. We call $\tilde{y}_n$ is *corrupted* if $\tilde{y}_n \neq y_n$ and clean otherwise. The instance $(x_n, \tilde{y}_n)$ is a *corrupted instance* if $\tilde{y}_n$ is corrupted. The noisy data distribution corresponds to $\widetilde{D}$ is $(X, \widetilde{Y}) \sim \widetilde{\mathcal{D}}$. We focus on the closed-set label noise that $Y$ and $\widetilde{Y}$ are assumed to be in the same label space, e.g., $Y, \widetilde{Y} \in [K]$ for a $K$-class classification task. Explorations on open-set data (Xia et al., 2020a; Luo et al., 2021) are deferred to future works.

**Representations** The *representation* of feature $x_n$ is denoted by $\bar{x}_n := g(x_n)$, where $g(\cdot)$ denotes a representation extractor. Feature $x_n$ could be in any shape and representation $\bar{x}_n$ is generally a high-dimensional vector. In this paper, we focus on a setting where the distances between two representations should be comparable or clusterable (Zhu et al., 2021b), i.e., nearby representations should belong to the same true class with a high probability (Gao et al., 2016), which could be defined as $(k, \delta_k)$ label clusterability in Definition 1.

**Definition 1** ($(k, \delta_k)$ label clusterability). *Given a representation extractor $g(\cdot)$, a dataset $D$ satisfies $(k, \delta_k)$ label clusterability if: $\forall n \in [N]$, the representation $\bar{x}_n := g(x_n)$ and its $k$-Nearest-Neighbors ($k$-NN) $\bar{x}_{n_1}, \cdots, \bar{x}_{n_k}$ belong to the same true class with probability at least $1 - \delta_k$.*

Note $\delta_k$ captures two types of randomnesses: one comes from a probabilistic $Y|X$, i.e., $\exists i \in [K], \mathbb{P}(Y = i|X) \notin \{0, 1\}$; the other depends on the quality of representation and the value of $k$, which will be further illustrated in Figure 2. The $(k, 0)$ label clusterability is also known as $k$-NN label clusterability (Zhu et al., 2021b). It has been shown in (Zhu et al., 2021b) that $k$-NN label clusterability can often be satisfied for a reasonable representation when $k$ is moderate to small. We define $k$ *perfect extractors/representations* in Definition 2.

**Definition 2** ($k$ perfect extractor/representation). *A representation extractor $g(\cdot)$ is called $k$ perfect extractor if it induces $k$-NN label clusterability. Its outputs are called $k$ perfect representations.*

**Label noise detection** Our paper aims to improve the performance of the label noise detection (a.k.a, detecting noisy/corrupted labels, finding label errors) which is measured by the $F_1$-score of the detected corrupted instances, which is the harmonic mean of the precision and recall, i.e. $F_1 = 2/(\texttt{Precision}^{-1} + \texttt{Recall}^{-1})$. Let $\mathbb{1}(\cdot)$ be the indicator function that takes value 1 when the specified condition is satisfied and 0 otherwise. Let $v_n = 1$ indicate that $\tilde{y}_n$ is detected as a corrupted label, and $v_n = 0$ if $\tilde{y}_n$ is detected to be clean. Then the precision and recall can be calculated as $\texttt{Precision} = \frac{\sum_{n\in[N]} \mathbb{1}(v_n=1, \tilde{y}_n \neq y_n)}{\sum_{n\in[N]} \mathbb{1}(v_n=1)}, \texttt{Recall} = \frac{\sum_{n\in[N]} \mathbb{1}(v_n=1, \tilde{y}_n \neq y_n)}{\sum_{n\in[N]} \mathbb{1}(\tilde{y}_n \neq y_n)}$.

## 3  LABEL NOISE DETECTION USING SIMILAR REPRESENTATIONS

Different from most methods that detect noisy labels based on the logit layer or model predictions (Northcutt et al., 2021a; Cheng et al., 2021; Pruthi et al., 2020), we focus on representations of features. Particularly we are interested in the possibility of detecting noisy labels in a training-free way, once given good representations. In this section, we will first introduce intuitions, and then provide two efficient algorithms to detect label noise with similar representations.

### 3.1  INTUITIONS

The training-based detection methods often make decisions by comparing model predictions with noisy labels (Cheng et al., 2021; Northcutt et al., 2021a). However, the representation $g(x_n)$ cannot be directly compared with the noisy label $\tilde{y}_n$ since $g(x_n)$ is not directly categorical, i.e., the output of a single $g(x_n)$ rarely corresponds to any label class. Thus the first step of our training-free solution should be establishing an auxiliary categorical information using only representations.

As illustrated in Figure 1, the high-level intuition is to check label consensuses of nearby representations. With $k$ perfect representations as defined in Definition 2, we know the true labels of $\bar{x}_n$ and its $k$-NN $\bar{x}_{n_1}, \cdots, \bar{x}_{n_k}$ should be the same. If the label noise across these instances are group-dependent (Wang et al., 2021), we can treat their noisy labels as $k + 1$ independent observations of $\mathbb{P}(\widetilde{Y}|X = x_n, Y = y_n)$, then estimate the probability by counting the frequency or weighted frequency[1] of each class in the $k$-NN *label estimator*, and get $k$-NN *labels* $\hat{\boldsymbol{y}}_n$. The $i$-th element $\hat{y}_n[i]$ can be interpreted as the estimated probability of predicting class-$i$.

**Representations could be better than model predictions** During supervised training, memorizing noisy labels makes the model generalizes poorly (Li et al., 2020a; Han et al., 2020), while using only representations may effectively avoid this issue. This is because representations often come from techniques beyond memorization such as unsupervised learning or self-supervised learning where the noisy labels are not referred in representation learning. On the other hand, the extractor $g(\cdot)$ has great potential in domain adaption or transfer, which makes representations more flexible than the specific model predictions. We will demonstrate this point in Section 5.

### 3.2 Voting-Based Local Detection

Inspired by the idea implemented in model decisions, i.e., selecting the most likely class as the true class, we can simply "predict" the index that corresponds to the largest element in $\hat{\boldsymbol{y}}_n$ with random tie-breaking, i.e., $y_n^{\texttt{vote}} = \arg\max_{i \in [K]} \hat{y}_n[i]$. To further detect whether $\tilde{y}_n$ is corrupted or not, we only need to check $v_n := \mathbb{1}(y_n^{\texttt{vote}} \neq \tilde{y}_n)$. Recall $v_n = 1$ indicates a corrupted label. This voting method relies only on the local information within each $k$-NN label $\hat{\boldsymbol{y}}_n$, which may not be robust with imperfect representations. Intuitively, when the gap between the true class probability and the wrong class probability is small, the majority vote will be likely to make mistakes due to sampling errors in $\hat{\boldsymbol{y}}_n$. Thus only using local information within each $\hat{\boldsymbol{y}}_n$ may not be sufficient. It is important to leverage more information such as some global statistics, which will be discussed later.

### 3.3 Ranking-Based Global Detection

From a global perspective, if the likelihood for each instance being clean could be evaluated by some scoring functions, we can sort the scores in an increasing order and filter out the low-score instances as corrupted ones. Based on this intuition, there are two critical components: the *scoring function* and the *threshold* to differentiate the low-score part (corrupted) and the high-score part (clean).

**Scoring function** A good scoring function should be able to give clean instances higher scores than corrupted instances. We adopt cosine similarity defined as: $\mathsf{Score}(\hat{\boldsymbol{y}}_n, j) = \frac{\hat{\boldsymbol{y}}_n^\top \boldsymbol{e}_j}{\|\hat{\boldsymbol{y}}_n\|_2 \|\boldsymbol{e}_j\|_2}$, where $\boldsymbol{e}_j$ is the one-hot encoding of label $j$. To evaluate whether the soft label $\hat{\boldsymbol{y}}_n$ informs us a clean instance or not, we compare $\mathsf{Score}(\hat{\boldsymbol{y}}_n, \tilde{y}_n)$ with other instances that have the same noisy label. This scoring function captures more information than majority votes, which is summarized as follows.

**Property 1** (Relative score). *Within the same instance, the score of the majority class is higher than the others, i.e.* $\mathsf{Score}(\hat{\boldsymbol{y}}_n, y_n^{\texttt{vote}}) > \mathsf{Score}(\hat{\boldsymbol{y}}_n, j), \forall j \neq y_n^{\texttt{vote}}, j \in [K], \forall n \in [N]$.

**Property 2** (Absolute score). $\mathsf{Score}(\hat{\boldsymbol{y}}_n, j)$ *is jointly determined by both* $\hat{y}_n[j]$ *and* $\hat{y}_n[j'], \forall j' \neq j$.

The first property guarantees that the corrupted labels would have lower scores than clean labels for the same instance when the vote is correct. However, although solely relying on Property 1 may work well in the voting-based method which makes decisions individually for each instance, it is not sufficient to be trustworthy in the ranking-based global detection. The main reason is that, across different instances, the non-majority classes of some instances may have higher absolute scores than the majority classes of the other instances, which is especially true for general instance-dependent label noise with heterogeneous noise rates (Cheng et al., 2021). Property 2 helps make it less likely to happen. Consider an example as follows.

*Example* Suppose $\hat{\boldsymbol{y}}_{n_1} = \hat{\boldsymbol{y}}_{n_2} = [0.6, 0.4, 0.0]^\top$, $\hat{\boldsymbol{y}}_{n_3} = [0.34, 0.33, 0.33]^\top$, $y_{n_1} = y_{n_2} = y_{n_3} = 1$, $\tilde{y}_{n_1} = \tilde{y}_{n_3} = 1, \tilde{y}_{n_2} = 2$. We can use the majority vote to get perfect detection in this case, i.e., $y_{n_1}^{\texttt{vote}} = y_{n_2}^{\texttt{vote}} = y_{n_3}^{\texttt{vote}} = 1 = y_{n_1}$, since the first class of each instance has the largest value. However, if we directly use a single value in soft label $\boldsymbol{y}_n$ to score them, e.g., $\mathsf{Score}'(\hat{\boldsymbol{y}}_n, j) =$

---

[1]One can weight instances by the similarity to the center instance.

---

**Algorithm 1** Detection with **Simi**lar **Rep**resentations (The SimiRep Detector)

---

1: **Input:** Number of epochs: $M$. $k$-NN parameter: $k$. Noisy dataset: $\widetilde{D} = \{(x_n, \tilde{y}_n)\}_{n\in[N]}$. Representation extractor: $g(\cdot)$. Method: *Vote* or *Rank*. Epoch counter $m = 0$.
2: **repeat**
3:    $x'_n \leftarrow \texttt{RandPreProcess}(x_n), \forall n;$          *# Initialize & Standard data augmentations*
4:    $\bar{x}_n \leftarrow g(x'_n), \forall n;$                        *# Extract representations with $g(\cdot)$*
5:    $\hat{y}_n \leftarrow \texttt{kNNLabel}(\{\bar{x}_n\}_{n\in[N]}, k)$    *# Get soft labels based on the clusterability of representations*
6:    **if** *Vote* **then**
7:       $y_n^{\text{vote}} \leftarrow \arg\max_{i\in[K]} \hat{y}_n[i];$                 *# Apply local majority vote*
8:       $v_n \leftarrow \mathbb{1}(y_n^{\text{vote}} \neq \tilde{y}_n), \forall n \in [N];$    *# Treat as corrupted if majority votes agree with noisy labels*
9:    **else**
10:      $\mathbb{P}(Y), \mathbb{P}(\widetilde{Y}|Y) \leftarrow \texttt{HOC}(\{(\bar{x}_n, \tilde{y}_n)\}_{n\in[N]});$
                              *# Estimate clean priors $\mathbb{P}(Y)$ and noise transitions $\mathbb{P}(\widetilde{Y}|Y)$ by the HOC estimator*
11:      $\mathbb{P}(Y|\widetilde{Y}) = \mathbb{P}(\widetilde{Y}|Y) \cdot \mathbb{P}(Y)/\mathbb{P}(\widetilde{Y});$          *# Estimate thresholds by Bayes' rule*
12:      **for** $j$ **in** $[K]$ **do**
13:         $\mathcal{N}_j := \{n|\tilde{y}_n = j\};$                  *# Detect corrupted labels in each set $\mathcal{N}_j$*
14:         $\mathcal{I} \leftarrow \texttt{argsort}\{\textsf{Score}(\hat{y}_n, j)\}_{n\in\mathcal{N}_j};$     *# $\mathcal{I}$ records the raw index of each sorted value*
15:         $v_n \leftarrow \mathbb{1}\big(\texttt{Loc}(n, \mathcal{I}) \leq \lfloor(1 - \mathbb{P}(Y = j|\widetilde{Y} = j)) \cdot N_j\rfloor\big);$
                                 *# Select low-score (head) instances as corrupted ones*
16:      **end for**
17:    **end if**
18:    $\mathcal{V}_m = \{v_n\}_{n\in[N]};$                      *# Record detection results in the $m$-th epoch*
19: **until** $M$ times
20: $\mathcal{V} = \texttt{Vote}(\mathcal{V}_m, \forall m \in [M]);$         *# Do majority vote based on results from $M$ epochs*
21: **Output:** $[N] \setminus \mathcal{V}$.

---

$\hat{y}_n[j]$, we will have $\hat{y}_{n_1}[\tilde{y}_{n_1}] = 0.6 > \hat{y}_{n_2}[\tilde{y}_{n_2}] = 0.4 > \hat{y}_{n_3}[\tilde{y}_{n_3}] = 0.33$, where the ranking is $n_3 \prec n_2 \prec n_1$. Ideally, we know instance $n_2$ is corrupted and the true ranking should be $n_2 \prec n_3 \prec n_1$ or $n_2 \prec n_1 \prec n_3$. To mitigate this problem, we choose the cosine similarity as our scoring function. The three instances could be scored as $0.83, 0.55, 0.59$, corresponding to an ideal ranking $n_2 \prec n_3 \prec n_1$. We formally introduce the detailed ranking approach as follows.

**Ranking** Suppose we have a group of instances with the same noisy class $j$, i.e. $\{(x_n, \tilde{y}_n)\}_{n\in\mathcal{N}_j}$, where $\mathcal{N}_j := \{n|\tilde{y}_n = j\}$ are the set of indices that correspond to noisy class $j$. Let $N_j$ be the number of indices in $\mathcal{N}_j$ (counted from noisy labels). Intuitively, we can first sort all instances in $\mathcal{N}_j$ in an increasing order by $\texttt{argsort}$ and obtain the original indices for the sorted scores as: $\mathcal{I} = \texttt{argsort}\{\textsf{Score}(\hat{y}_n, j)\}_{n\in\mathcal{N}_j}$, where the low-score head is supposed to consist of corrupted instances (Northcutt et al., 2021a). Then we can simply select the first $\widetilde{N}_j$ instances with low scores as corrupted instances: $v_n = \mathbb{1}(\texttt{Loc}(n, \mathcal{I}) \leq \widetilde{N}_j)$, where $\texttt{Loc}(n, \mathcal{I})$ returns the index of $n$ in $\mathcal{I}$. Instead of manually tuning $\widetilde{N}_j$, we discuss how to determine it algorithmically.

**Threshold** The number of corrupted instances in $\mathcal{N}_j$ is approximately $\mathbb{P}(Y \neq j|\widetilde{Y} = j) \cdot N_j$ when $N_j$ is sufficiently large. Therefore if all the corrupted instances have lower scores than any clean instance, we can set $\widetilde{N}_j = \mathbb{P}(Y \neq j|\widetilde{Y} = j) \cdot N_j$ to obtain the ideal division. Note $N_j$ can be obtained by directing counting the number of instances with noisy label $j$. To calculate the probability $\mathbb{P}(Y \neq j|\widetilde{Y} = j) = 1 - \mathbb{P}(Y = j|\widetilde{Y} = j)$, we borrow the results from the HOC estimator (Zhu et al., 2021b), where the noise transition probability $\mathbb{P}(\widetilde{Y} = j|Y = j)$ and the marginal distribution of clean label $\mathbb{P}(Y = j)$ can be estimated with only representations and the corresponding noisy labels. Then we can calculated our needed probability by Bayes' rule $\mathbb{P}(Y = j|\widetilde{Y} = j) = \mathbb{P}(\widetilde{Y} = j|Y = j) \cdot \mathbb{P}(Y = j)/\mathbb{P}(\widetilde{Y} = j)$, where $\mathbb{P}(\widetilde{Y} = j)$ can be estimated by counting the frequency of noisy label $j$ in $\widetilde{D}$. Technically other methods exist in the literature to estimate $\mathbb{P}(\widetilde{Y}|Y)$ (Liu & Tao, 2015; Patrini et al., 2017; Northcutt et al., 2021a; Li et al., 2021). But they often require training a model to fit the data distribution, which conflict with our goal of a training-free solution; instead, HOC fits us perfectly.

### 3.4 Algorithm: Detection with Similar Representations (SimiRep)

Algorithm 1 summarizes our solution. The main computation complexity is obtaining representations with extractor $g(\cdot)$, which is less than the cost of evaluating the model compared with the training-based methods. Thus SimiRep can filter out corrupted instances efficiently. In Algorithm 1, we run either voting-based local detection as Lines 7, 8, or ranking-based global detection as Lines 14, 15. The detection is run multiple times with random standard data augmentations to reduce the variance of estimation. The majority of results from different epochs is adopted as the final detection output as Line 20, i.e., flag as corrupted if $v_n = 1$ in more than half of the epochs.

## 4 How Do Representations Affect Our Solution?

In this section, we will first show how a good but imperfect representation[2] affects the selection of the hyperparameter $k$. We then offer an analysis of the error upper bound for the ranking-based method given a representation extractor $g(\cdot)$.

### 4.1 How Do Representations Affect the Choice of $k$?

Recall $k$ is used as illustrated in Figure 1. On one hand, the $k$-NN label estimator will be more accurate if there is stronger clusterability that more neighbor representations belong to the same true class (Liu & Liu, 2015; Zhu et al., 2021b), which helps improve the performance of later algorithms. On the other hand, with good but imperfect representations, stronger clusterability with a larger $k$ is less likely to satisfy, thus the violation probability $\delta_k$ increases with $k$ for a given extractor $g(\cdot)$. We take the voting-based method as an example and analyze this tradeoff. For a clean presentation, we focus on a binary classification with instance-dependent label noise where $\mathbb{P}(Y = 1) = p$, $\mathbb{P}(\widetilde{Y} = 2|X, Y = 1) = e_1(X), \mathbb{P}(\widetilde{Y} = 1|X, Y = 2) = e_2(X)$. Suppose the instance-dependent noise rate is upper-bounded by $e$, i.e., $e_1(X) \leq e, e_2(X) \leq e$. With $\delta_k$ as in Definition 1, we calculate the lower bound of the probability that the vote is correct in Proposition 1.

**Proposition 1.** *The lower bound for the probability of getting true detection with majority vote is*

$$\mathbb{P}(\textit{Vote is correct}|k) \geq (1 - \delta_k) \cdot I_{1-e_1}(k + 1 - k', k' + 1),$$

*where $k' = \lceil (k+1)/2 \rceil - 1$, $I_{1-e_1}(k + 1 - k', k' + 1)$ is the regularized incomplete beta function defined as $I_{1-e}(k + 1 - k', k' + 1) = (k + 1 - k')\binom{k+1}{k'}\int_0^{1-e} t^{k-k'}(1 - t)^{k'} dt$.*

Proposition 1 shows the tradeoff between a reliable $k$-NN label and an accurate vote. When $k$ is increasing, **Term-1** $(1 - \delta_k)$ (quality of representations) decreases but **Term-2** $I_{1-e_1}(k + 1 - k', k' + 1)$ (result of pure majority vote) increases. With Proposition 1, we are ready to answer the question: *when do we need more labels?* See Remark 1.

**Remark 1.** *Consider the lower bounds with $k_1$ and $k_2$ ($k_1 < k_2$). Supposing the first lower bound is lower than the second lower bound, based on Proposition 1, we roughly study the trend with an increasing $k$ by comparing two bounds and get*

$$\frac{1 - \delta_{k_1}}{1 - \delta_{k_2}} < \frac{I_{1-e}(k_2 + 1 - k_2', k_2' + 1)}{I_{1-e}(k_1 + 1 - k_1', k_1' + 1)}.$$

*For example, when $k_1 = 5$, $k_2 = 20$, $e = 0.4$, we can calculate the incomplete beta function and $\frac{1-\delta_5}{1-\delta_{20}} < 1.52$. Supposing $\delta_5 = 0.2$, we have $\delta_{20} < 0.47$. This indicates increasing $k$ from 5 to 20 would not improve the lower bound with imperfect representations with $\delta_{20} > 0.47$. This observation helps us set $k$ with a practical and imperfect $g(\cdot)$. We set $k = 10$ in all of our experiments.*

Remark 1 indicates that: ***with imperfect representations, a small $k$ may achieve the best (highest) probability lower bound***. To further consolidate this claim, we numerically calculate $\delta_k$ with different $g(\cdot)$ and the corresponding probability lower bound in Figure 2. We find most of the probability lower bounds first increase then decrease except for the perfect $g(\cdot)$ which is trained using ground-truth labels. Note the perfect $g(\cdot)$ has memorized all clean instances so that $\delta_k \to 0$ since $k \ll 5000$ (the number of instances in the same label class).

---

[2]Note the voting-based method achieves an $F_1$-score of 1 given $k$ perfect representations, $k \to +\infty$.

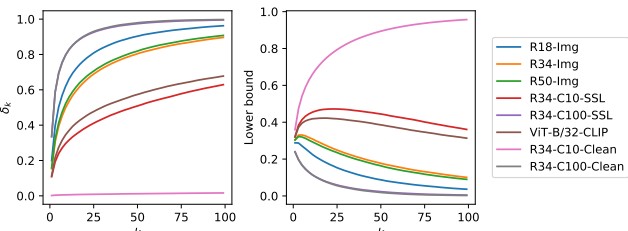

Figure 2: The trends of $\delta_k$ and probability lower bounds on CIFAR-10 (Krizhevsky et al., 2009) with different $g(\cdot)$. The outputs of the last convolution layer are adopted. R18/34/50: ResNet18/34/50. Img: Pre-trained on ImageNet (Deng et al., 2009). C10/100-Clean: Pre-trained on clean CIFAR-10/100. C10/100-SSL: Pre-trained on CIFAR-10/100 without labels by SimCLR (Chen et al., 2020). ViT-B/32-CLIP: CLIP Pre-trained vision transformer (Radford et al., 2021).

## 4.2 HOW DO REPRESENTATIONS AFFECT $F_1$-SCORE?

We next prove the probability bound for the performance of the ranking-based method. Consider a $K$-class classification problem with informative instance-dependent label noise (Cheng et al., 2021). Denote random variable $S$ by the score of each instance being clean. A higher score $S$ indicates the instance is more likely to be clean. Then for instances in $\mathcal{N}_j$, we have two set of random variables $\mathcal{S}_j^{\text{true}} := \{S^{\text{true}}|n \in \mathcal{N}_j, \tilde{y}_n = y_n\}$ and $\mathcal{S}_j^{\text{false}} := \{S^{\text{false}}|n \in \mathcal{N}_j, \tilde{y}_n \neq y_n\}$. Intuitively, the score $S_j^{\text{true}}$ should be greater than $S_j^{\text{false}}$. Suppose their means, which depend on noise rates, are bounded, i.e., $\mathbb{E}[S_j^{\text{true}}] \geq \mu_j^{\text{true}}$, $\mathbb{E}[S_j^{\text{false}}] \leq \mu_j^{\text{false}}$. Assume there exists a feasible $v$ such that both $S_j^{\text{true}}$ and $S_j^{\text{false}}$ follow sub-Gaussian distributions with variance proxy $\frac{\Delta^2}{2v}$ (Buldygin & Kozachenko, 1980; Zhu et al., 2021c) such that: $\mathbb{P}(\mu_j^{\text{true}} - S_j^{\text{true}} \geq t) \leq e^{-v(t/\Delta)^2}$, and $\mathbb{P}(S_j^{\text{false}} - \mu_j^{\text{false}} \geq t) \leq e^{-v(t/\Delta)^2}$, where $1/\Delta$ is the "height" of both distributions, i.e., $\mathbb{P}(S_j^{\text{true}} = \mu_j^{\text{true}}) = \mathbb{P}(S_j^{\text{false}} = \mu_j^{\text{false}}) = 1/\Delta$, $v$ is the decay rate of tails. Let $N_j^-$ ($N_j^+$) be the number of indices in $\mathcal{S}_j^{\text{false}}$ ($\mathcal{S}_j^{\text{true}}$). Theorem 1 summarizes the performance bound of the ranking-based method. See Appendix for the proof.

**Theorem 1.** *With probability at least $p$, the $F_1$-score of detecting corrupted instances in $\mathcal{N}_j$ with the rank-based method by threshold $N_j\mathbb{P}(Y = j|\widetilde{Y} = j)$ is at least $1 - \frac{e^{-v}\max(N^-,N^+)+\alpha}{N^-}$, where $p = \int_{-1}^{\mu^{true}-\mu^{false}-\Delta} f(t)dt$, $f(t)$ is the probability density function of the difference of two independent beta-distributed random variables $\beta_1 - \beta_2$, where $\beta_1 \sim Beta(N^-, 1), \beta_2 \sim Beta(\alpha+1, N^+ - \alpha)$.*

Theorem 1 shows the performance of detection depends on 1) the concentration of $S_j^{\text{true}}$ and $S_j^{\text{false}}$ (denoted by variance proxy $\frac{\Delta^2}{2v}$); 2) the distance between $S_j^{\text{true}}$ and $S_j^{\text{false}}$ (denoted by $\mu^{\text{true}} - \mu^{\text{false}}$). Intuitively, with proper scoring function and good representations, we have small variance proxy (small $\Delta$ and large $v$) and $F_1$-score approximates to 1.

## 5 EMPIRICAL RESULTS

We present experimental evidence in this section. The performance is measured by the $F_1$-score of the detected corrupted labels as defined in Section 2. Note there is no any training procedure in our method. The only hyperparameters in our methods are the number of epochs $M$ and the $k$-NN parameter $k$. Intuitively, a larger $M$ returns a collective result from more times of detection, which should be more accurate. The hyperparameter $k$ cannot be set too large as demonstrated in Figure 2. In CIFAR (Krizhevsky et al., 2009) experiments, rather than fine-tune $M$ and $k$ for different settings, we fix $M = 21$ and $k = 10$. We also test on Clothing1M (Xiao et al., 2015). Detailed experiment settings on Clothing1M are in Appendix C.

**Synthetic label noise** We experiment with three popular synthetic label noise models: the *symmetric* label noise, the *asymmetric* label noise, and the *instance-dependent* label noise. Denote the ratio of instances with corrupted labels in the whole dataset by $\eta$. Both the symmetric and the asymmetric noise models follow the class-dependent assumption (Liu & Tao, 2015), i.e., the label noise only depends only on the clean class: $\mathbb{P}(\widetilde{Y}|X, Y) = \mathbb{P}(\widetilde{Y}|Y)$. Specially, the symmetric noise is generated by uniform flipping, i.e., randomly flipping a true label to the other possible classes w.p. $\eta$ (Cheng et al., 2021). The asymmetric noise is generated by pair-wise flipping, i.e., randomly

Table 1: Comparisons of $F_1$-scores (%). CORES, CL, TracIn: Train with noisy supervisions. SimiRep-V and SimiRep-R: Get $g(\cdot)$ without any supervision. Top 2 are **bold**.

| Method | CIFAR10 | | | | CIFAR100 | | | |
|---|---|---|---|---|---|---|---|---|
| | Human | Symm. 0.6 | Asym. 0.3 | Inst. 0.4 | Human | Symm. 0.6 | Asym. 0.3 | Inst. 0.4 |
| CORES | 65.00 | 92.94 | 7.68 | **87.43** | 3.52 | **92.34** | 0.02 | 9.67 |
| CL | 55.85 | 80.59 | 76.45 | 62.89 | 64.58 | 78.98 | 52.96 | 50.08 |
| TracIn | 55.02 | 76.94 | 73.47 | 58.85 | 61.75 | 76.74 | 48.42 | 49.89 |
| SimiRep-V | **82.30** | **93.21** | **82.52** | 81.09 | **73.19** | 84.48 | **65.42** | **74.26** |
| SimiRep-R | **83.28** | **95.56** | **83.58** | 82.26 | **74.67** | **88.68** | 62.89 | 73.53 |

Table 2: Comparisons of $F_1$-scores (%). CORES, CL, TracIn: Use logit layers. SimiRep-V/R: Use only representations. All methods use the same extractor from CLIP. Top 2 are **bold**.

| Method | CIFAR10 | | | | CIFAR100 | | | |
|---|---|---|---|---|---|---|---|---|
| | Human | Symm. 0.6 | Asym. 0.3 | Inst. 0.4 | Human | Symm. 0.6 | Asym. 0.3 | Inst. 0.4 |
| CE Sieve | 67.21 | 94.56 | 5.24 | 8.41 | 16.24 | 88.55 | 2.6 | 1.63 |
| CORES | 83.18 | **96.94** | 12.05 | **88.89** | 38.52 | **92.33** | 7.02 | **85.52** |
| CL | 69.76 | 95.03 | 77.14 | 62.91 | 67.64 | 85.67 | 62.58 | 61.53 |
| TracIn | 81.85 | 95.96 | 80.75 | 64.97 | **79.32** | **91.03** | 63.12 | 64.31 |
| SimiRep-V | **87.43** | 96.44 | **88.97** | 87.11 | 76.26 | 86.88 | **73.50** | **80.03** |
| SimiRep-R | **87.45** | **96.74** | **89.04** | **91.14** | **79.21** | 90.54 | 68.14 | 77.37 |

flipping true label $i$ to the next class $(i \mod K) + 1$. Denote by $d$ the dimension of features. The instance-dependent label noise is synthesized by randomly generating a $d \times K$ projection matrix $w_i$ for each class $i$ and project each incoming feature with true class $y_n$ onto each column of $w_{y_n}$ (Xia et al., 2020b). Instance $n$ is more likely to be flipped to class $j$ if the projection value of $x_n$ on the $j$-th column of $w_{y_n}$ is high. See Appendix B in Xia et al. (2020b) and Appendix D.1 in Zhu et al. (2021b) for more details. We use symmetric noise with $\eta = 0.6$ (*Symm. 0.6*), asymmetric noise with $\eta = 0.3$ (*Asym. 0.3*), and instance-dependent noise with $\eta = 0.4$ (*Inst. 0.4*) in experiments.

**Real-world label noise** The *real-world* label noise comes from human annotations or weakly labeled web data. We use the $50,000$ noisy training labels ($\eta \approx 0.16$) for CIFAR-10 collected by Zhu et al. (2021b), and our self-collected $50,000$ noisy training labels ($\eta \approx 0.40$) for CIFAR-100 (data will be released in the non-anonymous version). Both sets of noisy labels are crowd-sourced from Amazon Mechanical Turk. For Clothing1M (Xiao et al., 2015), we could not calculate the $F_1$-scores due to the lack of ground-truth labels. We firstly perform noise detection on 1 million noisy training instances then train only with the selected clean data to check the effectiveness.

## 5.1 SUPERVISION MAY NOT BE NECESSARY

Our first experiment aims to show that training with noisy supervisions may not be necessary in detecting corrupted labels. To this end, we compare our methods, i.e., voting-based local detection (SimiRep-V) and ranking-based global detection (SimiRep-R), with three recent noise detection works: CORES (Cheng et al., 2021), confident learning (CL) (Northcutt et al., 2021a), and TracIn (Pruthi et al., 2020). We use ResNet34 as the backbone network in this experiment.

**Baseline settings** All these three baselines require training a model with the noisy supervision. Specifically, CORES (Cheng et al., 2021) trains ResNet34 on the noisy dataset and uses its proposed sample sieve to filter out the corrupted instances. We adopt its default setting during training and calculate the $F_1$-score of the sieved out corrupted instances. Confident learning (CL) (Northcutt et al., 2021a) detects corrupted labels by firstly estimating probabilistic thresholds to characterize label noise, ranking instances based on model predictions, then filtering out corrupted instances based on ranking and thresholds. We adopt its default hyper-parameter setting to train ResNet34. TracIn (Pruthi et al., 2020) detects corrupted labels by evaluating the self-influence of each instance, where the corrupted instances tend to have a high influence score. The influence scores are calculated based on gradients of the last layer of ResNet34 at epoch $40, 50, 60, 100$, where the model is trained with a batch size of $128$. The initial learning rate is $0.1$ and decays to $0.01$ at epoch $50$. Note TracIn only provides ranking for instances. To exactly detect corrupted instances, thresholds are required. For a fair comparison, we refer to the thresholds learned by confident learning (Northcutt et al., 2021a). Thus the corrupted instances selected by TracIn are based on the ranking from its self-influence and thresholds from CL. To highlight that our solutions work well without any supervision, our representation extractor $g(\cdot)$ comes from the ResNet34 pre-trained by SimCLR (Chen et al., 2020) where contrastive learning is applied and *no supervision* is required. Extractor $g(\cdot)$ is obtained with only in-distribution features, e.g., for experiments with CIFAR-10, $g(\cdot)$ is pre-trained with features only from CIFAR-10.

**Performance** Table 1 compares the results obtained with or without supervisions. We can see both the voting-based and the ranking-based method achieve overall higher $F_1$-scores compared with the

Table 4: Experiments on Clothing1M. None: Standard training with 1M noisy data. R50-Img (or ViT-B/32-CLIP, R50-Img Warmup-1): Apply our method with ResNet50 pre-trained on ImageNet (or ViT-B/32 pre-trained by CLIP, R50-Img with 1-epoch warmup). Top-1 is **bold**.

| Data Selection | # Training Samples | Best Epoch | Last 10 Epochs | Last Epoch |
|---|---|---|---|---|
| None (Standard Baseline) | 1M (100%) | 70.32 | $69.44 \pm 0.13$ | 69.53 |
| R50-Img | 770k (77.0%) | 72.37 | $71.95 \pm 0.08$ | 71.89 |
| ViT-B/32-CLIP | 700k (70.0%) | 72.54 | $72.23 \pm 0.17$ | 72.11 |
| R50-Img Warmup-1 | 767k (76.7%) | **73.64** | $\mathbf{73.28 \pm 0.18}$ | **73.41** |

other three results that require learning with noisy supervisions. Moreover, in detecting the real-world human-level noisy labels, our solution outperforms baselines around 20% on CIFAR-10 and 10% on CIFAR-100, which indicates the training-free solution are more robust to complicated noise patterns. One might also note that CORES achieves exceptionally low $F_1$-scores on CIFAR-10/100 with asymmetric noise and CIFAR-100 with human noise. This observation also informs us that customized training processes might not always be universally applicable.

## 5.2 REPRESENTATION LAYERS MAY PERFORM BETTER THAN LOGIT LAYERS

Table 2 shows the detection using either representation layers or logit layers of DNNs. In addition to the baselines compared in Section 5.1, we also compare to CE Sieve (Cheng et al., 2021) which follows the same sieving process as CORES but uses CE loss without regularizer. All methods adopt ViT-B/32 pre-trained by CLIP (Radford et al., 2021). Specifically, the training-based methods further fine-tune a linear logit layer on noisy supervisions, while our solution directly sets this pre-trained model as $g(\cdot)$. Other settings are the same as those in Section 5.1. By counting the frequency of reaching top-2 $F_1$-scores, we find SimiRep-R wins 1st place, SimiRep-V and CORES are tied for 2nd place. However, similar to Table 2, we find the training process of CORES to be unstable. For instance, it almost fails for CIFAR-100 with asymmetric noise. It is therefore reasonable to believe both methods with only representations achieve an overall higher $F_1$-score than other methods with logit layers. In other words, compared with our methods, the extra fine-tuning of logit layers with noisy supervisions cannot always help improve the performance of detecting corrupted labels.

## 5.3 WHAT ARE GOOD REPRESENTATIONS

Previous experiments demonstrate our methods overall outperform baselines given representations from contrastive pre-training. It is interesting to see how other representations perform. We summarize results of SimiRep-R in Table 3. There are several interesting findings: 1) The ImageNet pre-trained models perform well, indicating

Table 3: Comparisons of $F_1$-scores (%) using $g(\cdot)$ with different $\delta_k$ (%). Model names are the same as Figure 2.

| Pre-trained Model | CIFAR10 | | | CIFAR100 | | |
|---|---|---|---|---|---|---|
| | $1 - \delta_k$ | Human | Inst. 0.4 | $1 - \delta_k$ | Human | Inst. 0.4 |
| R18-Img | 35.73 | 75.40 | 80.22 | 11.30 | 74.91 | 71.99 |
| R34-Img | 48.13 | 79.52 | 82.43 | 16.17 | 76.88 | 74.00 |
| R50-Img | 45.77 | 78.40 | 82.06 | 15.81 | 76.55 | 73.51 |
| ViT-B/32-CLIP | 64.12 | 87.45 | 91.14 | 19.94 | 79.21 | 77.37 |
| R34-C10-SSL | 69.31 | 83.28 | 85.26 | 2.59 | 68.03 | 65.94 |
| R34-C10-Clean | 99.41 | 98.39 | 98.59 | 0.22 | 60.90 | 60.73 |
| R34-C100-SSL | 18.59 | 59.96 | 74.99 | 22.46 | 74.67 | 73.53 |
| R34-C100-Clean | 18.58 | 60.17 | 76.41 | 89.07 | 92.87 | 95.29 |

*the traditional supervised training on out-of-distribution data helps*; 2) For CIFAR-100, extractor $g(\cdot)$ obtained with only features from CIFAR-10 (R34-C10-SSL) performs better than the extractor with clean CIFAR-10 (R34-C10-Clean), indicating that *contrastive pre-training has better generalization ability to out-of-distribution data than supervised learning*; 3) The $F_1$-scores achieved by $g(\cdot)$ trained with the corresponding clean dataset are close to 1, indicating *our solution can give perfect detection with ideal representations*. Besides, we test the performance of training only with the clean instances selected by our approach in Table 4. Standard training with Cross-Entropy loss is adopted. The only difference between the first row and other rows of Table 4 is that some training instances are filtered out by our approach. Table 4 shows simply filtering out corrupted instances based on our approach distinctively outperforms the baseline. We also observe that slightly tuning $g(\cdot)$ in the fine-grained Clothing1M dataset would be helpful. See more details in Appendix C.

## 6 CONCLUSIONS

This paper proposed a new and universally applicable training-free solution to detect noisy labels by using the neighborhood information defined by a good set of representations. We have also demonstrated that good representations are reasonable assumptions with pre-training or self-supervised learning. Future works will explore other tasks that could benefit from label cleaning.

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

The omitted proofs are provided as follows.

## A    THEORETICAL ANALYSES

### A.1    PROOF FOR PROPOSITION 1

Now we derive a lower bound for the probability of getting true detection with majority vote:

$$
\begin{aligned}
\mathbb{P}(\text{Vote is correct}|k) \geq &(1 - \delta_k) \cdot \Bigg[ p \sum_{l=0}^{\lceil (k+1)/2 \rceil - 1} \binom{k+1}{l} e_1^l (1 - e_1)^{k+1-l} \\
&+ (1 - p) \sum_{l=0}^{\lceil (k+1)/2 \rceil - 1} \binom{k+1}{l} e_2^l (1 - e_2)^{k+1-l} \Bigg] \\
= &(1 - \delta_k) \cdot [p \cdot I_{1-e_1}(k + 1 - k', k' + 1) + (1 - p) \cdot I_{1-e_2}(k + 1 - k', k' + 1)]
\end{aligned}
$$

where $I_{1-e_1}(k + 1 - k', k' + 1)$ is the regularized incomplete beta function defined as

$$
I_{1-e}(k + 1 - k', k' + 1) = (k + 1 - k')\binom{k+1}{k'} \int_0^{1-e} t^{k-k'}(1 - t)^{k'} dt,
$$

and $k' = \lceil (k + 1)/2 \rceil - 1$.

## B    PROOF FOR THEOREM 1

*Proof.* Now we derive the worst-case error bound. We first repeat the notations defined in Section 4.2 as follows.

Denote random variable $S$ by the score of each instance being clean. A higher score $S$ indicates the instance is more likely to be clean. Then for instances in $\mathcal{N}_j$, we have two set of random variables $\mathcal{S}_j^{\text{true}} := \{S^{\text{true}} | n \in \mathcal{N}_j, \tilde{y}_n = y_n\}$ and $\mathcal{S}_j^{\text{false}} := \{S^{\text{false}} | n \in \mathcal{N}_j, \tilde{y}_n \neq y_n\}$. Intuitively, the score $S_j^{\text{true}}$ should be greater than $S_j^{\text{false}}$. Suppose their means, which depend on noise rates, are bounded, i.e., $\mathbb{E}[S_j^{\text{true}}] \geq \mu_j^{\text{true}}$, $\mathbb{E}[S_j^{\text{false}}] \leq \mu_j^{\text{false}}$. Assume there exists a feasible $v$ such that both $S_j^{\text{true}}$ and $S_j^{\text{false}}$ follow sub-Gaussian distributions with variance proxy $\frac{\Delta^2}{2v}$ (Buldygin & Kozachenko, 1980; Zhu et al., 2021c) such that: $\mathbb{P}(\mu_j^{\text{true}} - S_j^{\text{true}} \geq t) \leq e^{-v(t/\Delta)^2}$, and $\mathbb{P}(S_j^{\text{false}} - \mu_j^{\text{false}} \geq t) \leq e^{-v(t/\Delta)^2}$, where $1/\Delta$ is the "height" of both distributions, i.e., $\mathbb{P}(S_j^{\text{true}} = \mu_j^{\text{true}}) = \mathbb{P}(S_j^{\text{false}} = \mu_j^{\text{false}}) = 1/\Delta$, $v$ is the decay rate of tails. Let $N_j^-$ ($N_j^+$) be the number of indices in $\mathcal{S}_j^{\text{false}}$ ($\mathcal{S}_j^{\text{true}}$). For ease of notations, we omit the subscript $j$ in this proof since the detection is performed on each $j$ individually.

Denote the order statistics of random variables in $\mathcal{S}^{\text{false}}$ by $S_{(1)}^{\text{false}}, \cdots S_{(N^-)}^{\text{false}}$, where $S_{(1)}^{\text{false}}$ is the smallest order statistic and $S_{(N^-)}^{\text{false}}$ is the largest order statistic. The following lemma motivates the performance of the rank-based method.

**Lemma 1.** *The $F_1$-score of detecting corrupted labels in $\mathcal{N}_j$ by the rank-based method will be no less than $1 - \alpha/N^-$ when the true probability $\mathbb{P}(Y = j|\widetilde{Y} = j)$ is known and $S_{(N^-)}^{false} < S_{(\alpha+1)}^{true}$.*

Lemma 1 connects the upper bound for the number of wrongly detected corrupted instances with order statistics. There are two cases that can cause detection errors:
**Case-1**:

$$
0 \leq \mu^{\text{true}} - S^{\text{true}} < \Delta \ \text{ and } \ 0 \leq S^{\text{false}} - \mu^{\text{false}} < \Delta : \text{at most } \alpha \text{ errors when } S_{(N^-)}^{\text{false}} < S_{(\alpha+1)}^{\text{true}}.
$$

and **Case-2**:

$$
\mu^{\text{true}} - S^{\text{true}} \geq \Delta \ \text{ or } \ S^{\text{false}} - \mu^{\text{false}} \geq \Delta : \text{at most } \max(N_-, N_+) \text{ errors}
$$

We analyze each case as follows.

**Case-1:**   When Case-1 holds, we have

$$\mathbb{P}(\mu^{\text{true}} - S^{\text{true}} = x) \leq 1/\Delta, x \in [0, \Delta]$$

and

$$\mathbb{P}(S^{\text{false}} - \mu^{\text{true}} = x) \leq 1/\Delta, x \in [0, \Delta].$$

The above two inequalities show that the left tail of $S^{\text{true}}$ and the right tail of $S^{\text{false}}$ can be upper bounded by uniform distributions. Denote the corresponding uniform distribution by $U^{\text{true}} \sim \mathsf{Unif}(\mu^{\text{true}} - \Delta, \mu^{\text{true}})$ and $U^{\text{false}} \sim \mathsf{Unif}(\mu^{\text{false}}, \mu^{\text{false}} + \Delta)$.

With true $\mathbb{P}(Y = j | \widetilde{Y} = j)$, the detection errors only exist in the cases when the left tail of $S^{\text{true}}$ and the right tail of $S^{\text{false}}$ are overlapped. When the tails are upper bounded by uniform distributions, we have

$$\mathbb{P}(S^{\text{false}}_{(N^-)} < S^{\text{true}}_{(\alpha+1)}) \geq \mathbb{P}(U^{\text{false}}_{(N^-)} < U^{\text{true}}_{(\alpha+1)})$$

$$= \mathbb{P}\left( \left[U^{\text{false}} - \mu^{\text{false}}\right]_{(N^-)} + \mu^{\text{false}} < \left[U^{\text{true}} - (\mu^{\text{true}} - \Delta)\right]_{(\alpha+1)} + (\mu^{\text{true}} - \Delta) \right)$$

$$= \mathbb{P}\left( \left[U^{\text{false}} - \mu^{\text{false}}\right]_{(N^-)} - \left[U^{\text{true}} - (\mu^{\text{true}} - \Delta)\right]_{(\alpha+1)} < \mu^{\text{true}} - \mu^{\text{false}} - \Delta \right).$$

Note

$$\left[U^{\text{false}} - \mu^{\text{false}}\right]_{(N^-)} \sim Beta(N^-, 1),$$

and

$$\left[U^{\text{true}} - (\mu^{\text{true}} - \Delta)\right]_{(\alpha+1)} \sim Beta(\alpha + 1, N^+ - \alpha),$$

where $Beta$ denotes the Beta distribution. Both variables are independent. Thus the PDF of the difference is

$$f(p) = \begin{cases} B(N^+ - \alpha, 1)p^{N^+ - \alpha}(1-p)^{\alpha+1}F(1, N^- + N^+, 1 - N^-; \alpha + 2; 1 - p, 1 - p^2)/A, & 0 < p \leq 1 \\ B(N^-, N^+ - \alpha)(-p)^{N^+ - \alpha}(1+p)^{N^- + N^+ - \alpha - 1}F(N^+ - \alpha, -\alpha, N^- + N^+; N^- + N^+ - \alpha; 1 - p^2, 1 + p)/A, & -1 \leq p < 0 \\ B(N^- + \alpha, N^+ - \alpha)/A, & p = 0, \end{cases}$$

where $A = B(N^-, 1)B(\alpha + 1, N^+ - \alpha)$, $B(a, b) = \int_0^1 t^{a-1}(1-t)^{b-1}dt$

$$F(a, b_1, b_2; c; x, y) = \frac{\Gamma(c)}{\Gamma(a)\Gamma(c-a)} \int_0^1 t^{a-1}(1-t)^{c-a-1}(1 - xt)^{-b_1}(1 - yt)^{-b_2}\, \mathrm{d}t.$$

Therefore, we have

$$\mathbb{P}(S^{\text{false}}_{(N^-)} < S^{\text{true}}_{(\alpha+1)}) \geq \int_{-1}^{\mu^{\text{true}} - \mu^{\text{false}} - \Delta} f(p)dp.$$

**Case-2**   The other part, we have no more than $e^{-v} \cdot \max(N^-, N^+)$ corrupted instances that may have higher scores than one clean instance.

**Wrap-up**   From the above analyses, we know, w.p. at least $\int_{-1}^{\mu^{\text{true}} - \mu^{\text{false}} - \Delta} f(p)dp$, there are at most $e^{-v} \max(N^-, N^+) + \alpha$ errors in detection corrupted instances. Note Precision = Recall if we detect with the best threshold $N_j \mathbb{P}(Y = j | \widetilde{Y} = j)$. Therefore, the corresponding $F_1$-score would be at least $1 - \frac{e^{-v} \max(N^-, N^+) + \alpha}{N^-}$.

$\square$

## C   EXPERIMENT SETTINGS ON CLOTHING1M

We firstly perform noise detection on 1 million noisy training instances then train only with the selected clean data to check the effectiveness. Particularly, in each epoch of the noisy detection, we use a batch size of 32 and sample 1,000 mini-batches from 1M training instances while ensuring the (noisy) labels are balanced. We repeat noisy detection for 600 epochs to ensure a full coverage of 1 million training instances. Parameter $k$ is set to 10.

Table 5: Experiments on Clothing1M (Xiao et al., 2015) **with or without balanced sampling**. None: Standard training with 1M noisy data. R50-Img (or ViT-B/32-CLIP, R50-Img Warmup-1): Apply our method with ResNet50 pre-trained on ImageNet (or ViT-B/32 pre-trained by CLIP, R50-Img with 1-epoch warmup).

| Data Selection | # Training Samples | Best Epoch | Last 10 Epochs | Last Epoch |
|---|---|---|---|---|
| None (Standard Baseline) (Unbalanced) | 1M (100%) | 70.32 | $69.44 \pm 0.13$ | 69.53 |
| None (Standard Baseline)   (Balanced) | 1M (100%) | 72.20 | $71.40 \pm 0.31$ | 71.22 |
| R50-Img (Unbalanced) | 770k (77.0%) | 72.37 | $71.95 \pm 0.08$ | 71.89 |
| R50-Img   (Balanced) | 770k (77.0%) | 72.42 | $72.06 \pm 0.16$ | 72.24 |
| ViT-B/32-CLIP (Unbalanced) | 700k (70.0%) | 72.54 | $72.23 \pm 0.17$ | 72.11 |
| ViT-B/32-CLIP   (Balanced) | 700k (70.0%) | 72.99 | $72.76 \pm 0.15$ | 72.91 |
| R50-Img Warmup-1 (Unbalanced) | 767k (76.7%) | **73.64** | $\mathbf{73.28 \pm 0.18}$ | **73.41** |
| R50-Img Warmup-1   (Balanced) | 767k (76.7%) | **73.97** | $\mathbf{73.37 \pm 0.03}$ | 73.35 |

**Representation Extractor:** We tested three different representation extractors in Table 4: R50-Img, ViT-B/32-CLIP, and R50-Img Warmup-1. The former two representation extractors are the same as the ones used in Table 3. Particularly, R50-Img means the representation extractor is the standard ResNet50 encoder (removing the last linear layer) pre-trained on ImageNet (Deng et al., 2009). ViT-B/32-CLIP indicates the representation extractor is a vision transformer pre-trained by CLIP (Radford et al., 2021). Noting that Clothing1M is a fine-grained dataset. To get better domain-specific fine-grained visual representations, we slightly train the ResNet50 pre-trained with ImageNet for one epoch, i.e., 1,000 mini-batches (batch size 32) randomly sampled from 1M training instances while ensuring the (noisy) labels are balanced. The learning rate is 0.002.

**Training with the selected clean instances:** Given the selected clean instances from our approach, we directly apply the Cross-Entropy loss to train a ResNet50 initialized by standard ImageNet pre-trained parameters. We **did not** apply any sophisticated training techniques, e.g., mixup (Zhang et al., 2018), dual networks (Li et al., 2020b; Han et al., 2018), loss-correction (Liu & Tao, 2015; Natarajan et al., 2013; Patrini et al., 2017), and robust loss functions (Liu & Guo, 2020; Cheng et al., 2021; Zhu et al., 2021a; Wei & Liu, 2021). We train the model for $80$ epochs with a batch size of 32. We sample $1,000$ mini-batches per epoch randomly selected from 1M training instances. Note Table 4 does not apply balanced sampling. Only the pure cross-entropy loss is applied. We also test the performance with balanced training, i.e., in each epoch, ensure the noisy labels from each class are balanced. Our approach can be consistently benefited by balanced training, and achieves an accuracy of 73.97 in the best epoch, outperforming many baselines such as HOC 73.39% (Zhu et al., 2021b), GCE+SimCLR 73.35% (Ghosh & Lan, 2021), CORES 73.24% (Cheng et al., 2021), GCE 69.75% (Zhang & Sabuncu, 2018). We believe the performance could be further improved by using some sophisticated training techniques mentioned above.

