# OpenReview forum: "A Good Representation Detects Noisy Labels"
_ICLR.cc/2022/Conference — ICLR 2022 Submitted_

### Official Review · Reviewer_1co7 · 2021-11-02

**Correctness:** 3
**Technical Novelty And Significance:** 3
**Empirical Novelty And Significance:** 2
**Recommendation:** 5
**Confidence:** 3

**Details Of Ethics Concerns:**

not any

**Main Review:**

The paper is well organized and easy to follow. The intuition of the paper, which lies in utilizing proper representation instead of training a deep inference model to avoid memorizing corrupted data, also sounds somewhat interesting and seem to work. Besises, the proposed method can detect label noise in the instancewise manner. However, my concerns mainly lie in the methodology and the experiment.

First, what is the specific definition and criterion of good representation emphasized in this paper? It seems that the prerequisite of the proposed method is “given good representation”. The authors have done some basic tests in section 5.3 (table 3) that indicates the representations lead to accurate inference result is a good representation. However, it is the post-inference analysis, is there any way to select a good representation extractor g(·) in advance? Besides, the dependence on good representation may become the bottleneck of the proposed method since visual representation itself is an important and challenging topic.

Second, this paper takes the corrupted CIFAR-10 and CIFAR-100 (corrupted by automatic labeling tool) as the real-world noisy dataset for experiments, which is not very convincing. The acknowledged noisy-label dataset such as Webvision and Cothing1M could be more helpful in verifying the effectiveness and universality of the proposed method. Though these datasets may not involve the ground truth labels, there are still many ways to achieve the quantitative comparison test. For example, you can filter the wrong labels detected by the proposed method, and compare the overall inference accuracy between the proposed method and the SOTAs.

In addition, though the authors claim that their method is universal, they adopt clustering methods like k-NN to generate the soft labels and resolve the discrimination task. I thus wonder if the increasing number of object categories may result in a significant drop in noise detection accuracy. That is also one of the reasons why I wonder about the convincingness of the datasets used in this paper.

There are also some minor concerns：
1. The related work is too brief to reflect the association and difference between the proposed method and the existing methods.
2. The ambiguous meaning of words. Eg, why do the authors call the representation extractor g(·)  k perfect extractor if it induces k-NN label clusterability in Definition 2? Why perfect?
3. There are some writing issues to fix, the authors may need to proofread the papers more carefully. eg. Property 1, "With in"  in the sentence " With in the same instance, bla bla".

**Summary Of The Paper:**

The paper proposes a training-free instancewise noise label detection method. The main motivation of this paper is the observation that deep models generalize poorly because memorizing noisy labels in supervised training while using only representations may avoid this issue.  The authors suppose a good representation extractor is given and generate the initial soft labels based on the clusterability of representations using kNN. Followed by that,  they perform a local voting and global ranking-based scoring system to detect the corrupted labels. The main contribution of this paper is: It introduces a representation-based method instead of directly training a deep model on the corrupted data, which is more efficient and may avoid overfitting noisy labels.


**Summary Of The Review:**

The paper proposes a training-free instancewise noise label detection method. It supposes a good representation extractor is given and generates the initial soft labels based on the clusterability of representations using kNN. Followed by that,  the authors perform a local voting and global ranking-based scoring system to detect the corrupted labels. In general, the paper is well written and easy to follow. The main idea of utilizing good representation instead of training a deep inference model to avoid memorizing corrupted data sounds interesting and seems to work on CIFAR10 and CIFAR100 datasets.  However, I have some concerns about the definition and criterion of good representation which is the prerequisite of the method. As the proposed method uses kNN to analyze the clusterability of representations, I also wonder the increase of object categories may cause a significant accuracy drop in discrimination. Besides, the author only perform experiments on CIFAR10 and CIFAR100 instead of the acknowledged noisy-label datasets, which is not convincing enough for me. There are also some minor issues in the paper to be taken care of.

---

> ### Author Response · Authors · 2021-11-18
> **Response to Reviewer 1co7 (part 1)**
>
> Thanks for reviewing our paper and providing comments that help us improve the paper. We also would like to thank you for suggesting a quantitative way to evaluate our method on Clothing1M. Following your comments, we do experiments on Clothing1M and. Our method can select $\sim 76.7\\%$ clean instances from 1M training data and standard training with cross-entropy on this subset achieves an accuracy of $73.64\\%$. Please see more details at **General Response to All Reviewers**.
>
> Other concerns are addressed below.
>
>
> **Question 1:** First, what is the specific definition and criterion of good representation emphasized in this paper?
>
> **Response 1:**
>
> Note that we have $(k,\delta)$ label clusterability (Definition 1) to evaluate the goodness of representations. Figure 2 and Table 3, as presented in our original submission, are serving the exact purpose. Recall that we have $(k,\delta)$ label clusterability (Definition 1) to evaluate the goodness of representations. In Figure 2, we evaluate this goodness on different representation extractors, e.g., ImageNet pre-trained models, CLIP, SSL models. Please check Figure 2 at the top of Page 7 and the corresponding explanations after Figure 2. We also empirically showed how different representation extractors affect our detection accuracy in Table 3. Note a higher $1-\delta_k$ indicates a better representation extractor.
>
> **Question 2:** Is there any way to select a good representation extractor g(·) in advance?
>
> **Response 2:**
>
> Evaluating the quality of a good representation extractor may require some clean data (calculating $\delta_k$ as Figure 2 following Definition 1).
> But we would like to highlight that our approach is relatively robust to the selection of representation extractors in the sense that the performance does not drop too much with a relatively worse extractor. For example, in Table 3, when $1-\delta_k$, i.e., the quality of extractor decreases from $64.12\\%$ (ViT-B/32-CLIP) to $35.73\\%$ (R18-Img), we find the performance decreases from $87.45\\%$ to $75.40\\%$ (on CIFAR-10 with Human noise) and from $91.14\\%$ to $80.22\\%$ (on CIFAR-10 with Inst 0.4 noise). This means about a $30\\%$ drop in the quality of extractor may lead to only a $10\\%$ drop in F-scores. Additionally, Table 3 shows there is no distinct phase transition.
> Besides. it may not be necessary to select only one representation extractor in our method. For example, in different epochs of Algorithm 1, we may apply different $g(\cdot)$ in Line 4. Our method may be benefited by using multiple representation extractors by ensembling them. It is interesting to study efficient methods to weight different representation extractors in the future.
>
> **Question 3:** The dependence on good representation may become the bottleneck of the proposed method since visual representation itself is an important and challenging topic.
>
> **Response 3:**
>
> We agree that getting a good representation is critical in detecting label noise, which is one important message we want to show throughout the paper. Moreover, we argue that detection with representations is even more important than current methods based on model predictions. Note current detection with model predictions needs not only a good representation but also a good logit layer. Therefore, our approach is effectively moving the bottleneck from the whole model predictions to representations, which can be seen as a relaxation such that recent advances in representation learning can be used.
> We acknowledge that visual representation itself is an important and challenging topic. We believe our work organically links representation learning with noisy learning.
>
> **Question 4:**
> This paper takes the corrupted CIFAR-10 and CIFAR-100 (corrupted by automatic labeling tool) as the real-world noisy dataset for experiments, which is not very convincing.
>
> **Response 4:**
>
> We want to highlight that we also tested real-world human annotations on CIFAR in Tables 1--3 (please see the column called *Human*). Note these labels are **not generated by automatic labeling tool**. They are collected from Amazon Mechanical Turk as mentioned in the last paragraph called **Real-world label noise** before Section 5.1.
>
> **Question 5:** Experiments on Clothing1M
>
> **Response 5:**
> Thanks for suggesting this quantitative comparison test method. Please see the experiments on Clothing1M at the response: **General Response to All Reviewers.**

---

> > ### Comment · Reviewer_1co7 · 2021-12-01
> > **Thanks for the rebuttal**
> >
> > Thanks for preparing the rebuttal and extra experimental results. However, I think the authors may have mistaken my point. What I wonder is that they avoid tests on the acknowledged large-scale dataset with noise labels (clothing 1M and Webvision) in the first place, not the label corrupting methods.  Unfortunately, the experimental results provided in the rebuttal on Clothing 1M still lack contrasts with the SOTAs and are not solid enough to validate the effectiveness of the proposed method. Besides, the authors' explanation of the dependence on good representation still cannot persuade me to change my point. Thus I would like to keep my original rating (5).

---

> > > ### Author Response · Authors · 2021-12-01
> > > **thanks for the additional comments: re SOTA**
> > >
> > > We appreciate hearing back from the reviewers regarding our rebuttals. We honestly do not understand why achieving SOTA on Clothing1M determines the technical merit of a paper: the ability to detect corrupted labels has a more profound impact rather on achieving high classification accuracy on the dataset. Practically speaking, once we know which labels are wrong, we can bring human annotators back to the loop to re-label; one can also use this knowledge to understand the error patterns from people so as to better design the collection procedure. Of course, one can use the detection to improve classification accuracy. But restricting to the last application only seems to be a rather narrow take.
> > >
> > > We understand people have different opinions, but we want to kindly point out that ICLR has made it clear not achieving SOTA should not constitute grounds for rejection, per https://iclr.cc/Conferences/2022/ReviewerGuide :
> > >
> > > "Q: If a submission does not achieve state-of-the-art results, is that grounds for rejection?
> > >
> > > A: No, a lack of state-of-the-art results does not by itself constitute grounds for rejection. Submissions bring value to the ICLR community when they convincingly demonstrate new, relevant, impactful, or insightful knowledge. Submissions can achieve this without achieving state-of-the-art results.
> > > "

---

> > > > ### Comment · Reviewer_1co7 · 2021-12-01
> > > > **Thanks for the quick reply**
> > > >
> > > > The final score is based on various reasons. I think I have made my points clear. Additionally, there seems to be no conflict of point between different reviewers.  I understand it may be hard for the authors to hear a different voice.

---

> > > > > ### Author Response · Authors · 2021-12-01
> > > > > **glad to know it's not because of SOTA**
> > > > >
> > > > > Research progress requires different voices so we are definitely open to hearing them. We are glad to know the recommendation is not because of not achieving SOTA.

---

> ### Author Response · Authors · 2021-11-18
> **Response to Reviewer 1co7 (part 2)**
>
> **Question 6:**
> I wonder if the increasing number of object categories may result in a significant drop in noise detection accuracy.
>
> **Response 6:**
>
> Our experiments on CIFAR-100 show our approach works well when there are 100 classes.
> For example, in Table 1, our methods can achieve an F-score of $74\\%$ on CIFAR-100 with real-world human annotation noise, which outperforms baselines: CORES 3.52%, CL 64.58%, TracIn 61.75%. Note the real-world human annotation noise on CIFAR-100 is really challenging and the overall noise rate is about 40\%. Compared with the results on CIFAR-10 with human noise (16\% noise rate), our method only has a $9\\%$ performance drop (83\% vs. 74\%). It is reasonable our method does not suffer a significant drop in detection accuracy.
> We are not able to test on larger datasets (such as WebVision) due to the limitation of the computational resources.
>
> **Question 7:**
> The related work is too brief to reflect the association and difference between the proposed method and the existing methods.
>
> **Response 7:**
>
> We have a detailed comparison with the most relevant works in the first three paragraphs of the Introduction. Our related work (Section 1.1) reviews the literature from three aspects: learning with noisy labels, pre-training or self-supervised learning, and label aggregation.
> We appreciate it if the reviewers can provide more related references.
>
> **Question 8:**  k perfect extractor
>
> **Response 8:**
>
> In Definition 1, we have $(k,\delta_k)$ label clusterability, where $\delta_k$ is a violation probability. As mentioned before Definition 2, when $\delta_k=0$, we have $k$ perfect representations showing one and its k-NN belong to the same true labe class w.p. 1.
>
> **Others** Thanks for pointing out typos.

---

### Official Review · Reviewer_fWAV · 2021-11-03

**Correctness:** 2
**Technical Novelty And Significance:** 2
**Empirical Novelty And Significance:** 2
**Recommendation:** 5
**Confidence:** 4

**Main Review:**

Strengths
* The idea of using a pre-trained model to help detect noisy samples is interesting. The training-free approach is also largely different from existing works in literature.
* The finding in Sec 5.3 is particularly interesting as well. Table 3 in this section shows that contrastive pre-training shows better generalization ability than supervised pre-training for the task being addressed here.

Weaknesses
* The major concern of this work is about the experimental setup. CIFAR10/100 has a clear class definition across only 10 or 100 classes. By using a well-trained model pre-trained on a large amount of data, the representations of the samples from CIFAR are already well separated. That is also the reason the proposed K-NN approach works well compared to learning-based methods that do not have the advantage of large-scale pre-training. To justify the effectiveness of the proposed method in practice, it is crucial to test the algorithm on more complex and larger datasets.
* The theoretical analyses are highly appreciated in the submission. However, the analyses are based on a strong assumption that nearby representations should belong to the same true class. This might be true for simple CIFAR10 with a balanced training sample across 10 classes. In reality, the training set is usually imbalanced and nearby samples could have completely different labels.
* Figure 1 can be improved. The green circle in the figure is not defined.
* Typo: page 5 y_n3=[0.34, 0.033, 0.33] -> [0.34, 0.33, 0.33]

**Summary Of The Paper:**

The authors propose a training-free approach to detect samples with noisy labels by leveraging representations learned from pre-trained models. The models can be obtained by either supervised or self-supervised pre-training. The authors then argue that samples in the pre-trained manifold should be closer if they share the same clean label, therefore one can use (a) local voting or (b) ranking methods to detect samples with corrupted labels. Experiments on CIFAR10/100 show improvement over other learning-based approaches (CORES, CL, TracIn).

**Summary Of The Review:**

Although I appreciate the idea of using pre-trained models to provide guidance in selecting corrupted samples. It is also important to ensure the proposed method works well in practice, especially for imbalance or more complex large-scale settings.

---

> ### Author Response · Authors · 2021-11-18
> **Response to Reviewer fWAV**
>
> Thanks for your comments and for raising practical concerns. We test on Clothing1M, which is a fine-grained imbalanced large-scale dataset with real-world label noise. Our method can select $\sim 76.7\\%$ clean instances from 1M training data and standard training with cross-entropy on this subset achieves an accuracy of $73.64\\%$. Please see more details at **General Response to All Reviewers**.
>
> Other concerns are addressed as follows.
>
> **Question 1:** CIFAR10/100 has a clear class definition across only 10 or 100 classes. By using a well-trained model pre-trained on a large amount of data, the representations of the samples from CIFAR are already well separated. That is also the reason the proposed K-NN approach works well compared to learning-based methods that do not have the advantage of large-scale pre-training.
>
> **Response 1:**
>
> We totally agree with the reviewer on the advantage of large-scale pre-training, which is also one motivation of our work. But our comparison is fair since we **did not test the case where only our approach is benefited by large-scale pre-training** . **Note experiments in Table 1 do not access extra training data and all the methods in Table 2 adopt ViT-B/32 pre-trained by CLIP.** In our initial submission, **for Table 1, at the bottom of Page 8**, we have "To highlight that our solutions work well without any supervision, our representation extractor $g(\cdot)$ comes from the ResNet34 pre-trained by SimCLR (Chen et al., 2020) where contrastive learning is applied and no supervision is required. Extractor $g(\cdot)$ is obtained with only in-distribution features, e.g., for experiments with CIFAR-10, $g(\cdot)$ is pre-trained with features only from CIFAR-10." **For Table 2, in Section 5.2**, we have "All methods adopt ViT-B/32 pre-trained by CLIP (Radford et al., 2021)."
>
>
> **Question 2:** To justify the effectiveness of the proposed method in practice, it is crucial to test the algorithm on more complex and larger datasets.
>
> **Response 2:**
> We did experiments on Clothing1M in the revised version. Please check the response: **General Response to All Reviewers.**
>
> **Question 3:** The theoretical analyses are highly appreciated in the submission. However, the analyses are based on a strong assumption that nearby representations should belong to the same true class. This might be true for simple CIFAR10 with a balanced training sample across 10 classes. In reality, the training set is usually imbalanced and nearby samples could have completely different labels.
>
> **Response 3:**
>
> Thanks for supporting our analyses! But our clusterability condition is not that strong especially when $k=10$ as tested in our experiment. We agree that the training set is usually imbalanced/long-tailed in practice. Note the imbalance exists in two aspects: clean labels and noisy labels. Our experiments have tested the imbalanced noisy labels cases in Tables 1, 2, 3 since *Human* and *Inst. 0.4* are imbalanced in terms of noisy labels. For the clean-imbalanced cases, nearby samples in the tail part may not belong to the same class when $k$ is large. But our experiments (also from Figure 2) show $k=10$ is a good choice, which means we do not rely on a very strong clusterability. It is reasonable to believe that $(k,\delta_k)$ label clusterability has an acceptable $\delta_k$ when $k=10$ in the imbalanced/long-tailed setting. We also tested imbalanced CIFAR-10 (5k samples in each of the first 5 classes, 500 samples in each of the other classes) with human noise. The results (F-scores) are as follows:
> * SimiRep-V: 90.52
> * SimiRep-R: 81.69
> * CORES:74.12.
>
> Note the gap between SimiRep-V and SimiRep-R maybe because the estimate of the transition matrix is not sufficiently stable. Besides, Clothing1M is also an imbalanced dataset. More details are available at **General Response to All Reviewers.** We acknowledge the challenge in dealing with long-tail distributions, which is an interesting future direction.
>
> **Others:** Thanks for showing typos and helping us improve the paper.

---

> > ### Comment · Reviewer_fWAV · 2021-11-26
> > **Thanks for the rebuttal**
> >
> > Thanks to the authors for preparing the rebuttal and additional results. However, the majority of the arguments in the submission are still based on the simple CIFAR10/100 experiments. The additional experiments on Clothing1M mainly explore different backbones/warmup strategies. It is still hard to justify whether the proposed algorithm re-defines good representations or is effective on large-scale and practical settings. I would like to keep my original rating (5).

---

> > > ### Author Response · Authors · 2021-11-27
> > > **Thanks for the comment**
> > >
> > > Dear Reviewer fWAV,
> > >
> > > Thank you for the further comment. We would like to highlight two points as follows.
> > >
> > > * **Definition of good representations:** We acknowledge that **representation learning itself is a very challenging topic in the literature, which is out of the scope of our paper**. Our focus is only on the processing of a noisy dataset **given** a particular representation extractor. We know the linear evaluation protocol is a standard approach to evaluate representations (where a linear classifier is trained given a fixed representation extractor, and test accuracy is used as a proxy for representation quality). But our definitions of good representations (e.g., Def. 1 & 2) are proposed to **help our theoretical analyses** and **do not conflict with** the widely used linear evaluation protocol in SSL. Intuitively, linear separable representations tend to have better $(k,\delta_k)$ label clusterability (smaller $\delta_k$ given $k$).
> > >
> > > * **Large-scale and practical settings:** We agree that obtaining good representations in practical settings is challenging. But again, it is out of the scope of our paper since we only care about the processing of datasets given representations. Our theoretical analyses in Section 4 (Proposition 1, Remark 1, Theorem 1) show how our method would perform given a particular extractor. Our experiments such as Tables 1,2,3 on CIFAR-10/100 (with both synthetic label noise and real-world label noise), and Table 4 on Clothing1M show the **label noise detection using only representations could achieve competitive performance**. Note experiments on CIFAR-10/100 and Clothing1M are two popular datasets and **well-accepted by the noisy-learning literature**. All these experiments are fair comparisons and well-support our claim. Note we are not able to test on larger datasets (such as WebVision) due to the limitation of the computational resources. We would appreciate it if the reviewer could provide more practical datasets that are commonly used by the noisy-learning community (also, please note we did not focus on obtaining representation extractors).
> > >
> > > Looking forward to your comments!
> > >
> > > Best,
> > >
> > > ICLR 2022 Conference Paper468 Authors

---

### Official Review · Reviewer_39dB · 2021-11-03

**Correctness:** 3
**Technical Novelty And Significance:** 2
**Empirical Novelty And Significance:** 3
**Recommendation:** 6
**Confidence:** 4

**Main Review:**

Strengths: the paper proposes a new perspective to detect noisy labels
Weaknesses: my main concern is on the good representation assumption. To be more specific,

1. Though it is possible to obtain some good representations via pre-training on other datasets or self-supervised learning on the noisy datasets, it is still unclear how the obtained representations could be guaranteed to help with noisy label detection. It would be helpful to include experiments evaluating the goodness of representations obtained from various approaches based on the proposed criteria (e.g., Definition 1 and 2).

2. In Table 1 and 2, as the proposed method uses representations pre-trained by SimCLR, a self-supervised learning method, it seems unfair to compare with other noise detection methods, which do not have access to the good representations from SSL.

3. The paper claims the proposed method as training-free, which may be misleading and a bit overclaim since the good representation assumed by the proposed method still requires learning. Supervision-free maybe a better choice.

**Summary Of The Paper:**

This paper proposes a method to detect noisy labels given good representations (e.g., ones pre-trained by contrastive learning approaches).
The proposed noisy-label detection method uses  the neighborhood information defined by a good set of representations in two ways:

1) checks the noisy label consensuses of nearby representations

2) scores each instance by its likelihood of being clean and filters out a guaranteed percentage of instances with low scores as corrupted ones

The work provide definitions for good representation and further proves a worst-case error bound for the ranking-based method given a 'good enough' representation. It also provides empirical results for its proposed method.

**Summary Of The Review:**

This work proposes an interesting perspective to analyze how the representations could help with noise detection. Yet, the major assumption of this work that a good representation is given still needs further justifications. Additional experimental results are also needed to fully support the claim.

===Update after rebuttal===
Thanks to the authors' response, which resolved some of my confusions. I am increasing my score for the additional discussions and results

---

> ### Author Response · Authors · 2021-11-18
> **Response to Reviewer 39dB**
>
> Thanks for providing these comments to help up improve the paper. We test our approach on Clothing1M in this revised version. Please see the response: **General Response to All Reviewers**. **In our initial submission, we had some analyses about the quality of representations and the detection performance**. Your concerns are addressed as follows.
>
>
> **Question 1:** It is still unclear how the obtained representations could be guaranteed to help with noisy label detection.
>
> **Response 1:**
>
> We proposed metrics to evaluate the quality of representations in Definition 1 and 2. Given good but imperfect representations, our approach is analyzed both theoretically (Section 4) and empirically (Section 5). Note getting good representations itself is admittedly a challenging problem (e.g., pre-training or self-supervised learning), and is enjoying arguably a wider audience, but is beyond the scope of this paper. Our focus is to prototype the idea of detecting corrupted labels, given good representations.
>
> **Question 2:** It would be helpful to include experiments evaluating the goodness of representations obtained from various approaches based on the proposed criteria (e.g., Definition 1 and 2).
>
> **Response 2:**
>
> Yes, we agree such evaluation would be helpful. **Figure 2 and Table 3, as presented in our original submission, are serving the exact purpose**. Recall that we have $(k,\delta)$ label clusterability (Definition 1) to evaluate the goodness of representations. In Figure 2, we evaluate this goodness on different representation extractors, e.g., ImageNet pre-trained models, CLIP, SSL models. Please check Figure 2 at the top of Page 7 and the corresponding explanations after Figure 2. We also empirically showed how different representation extractors affect our detection accuracy in Table 3. Note a higher $1-\delta_k$ indicates a better representation extractor.
>
> **Question 3:** In Table 1 and 2, as the proposed method uses representations pre-trained by SimCLR, a self-supervised learning method, it seems unfair to compare with other noise detection methods, which do not have access to the good representations from SSL.
>
> **Response 3:**
>
> This seems to be a misunderstanding. In Table 2, as we mentioned in the 4-th line of Section 5.2 in our initial submission, **All methods adopt ViT-B/32 pre-trained by CLIP (Radford et al., 2021)**. Namely, **all the methods have access to the same good representations**. Note Table 1 and Table 2 aim to show different messages. We agree that SSL may provide good representations. Although in Table 1 we use SimCLR to get visual representations for CIFAR, it is still a fair comparison in terms of justifying our aim that training with noisy supervisions may not always be necessary for detecting corrupted labels.
>
> **Question 4:** The paper claims the proposed method as training-free, which may be misleading and a bit overclaim since the good representation assumed by the proposed method still requires learning. Supervision-free maybe a better choice.
>
> **Response 4:**
>
> We want to use training-free to show our method does not need any training procedures given good representations. We also considered calling it supervision-free but we do **need the noisy supervisions when we detect noisy labels**, which are exactly the information we need to clean up. We will further highlight the meaning of "training-free" to avoid confusion.

---

### Official Review · Reviewer_fRLd · 2021-11-04

**Correctness:** 3
**Technical Novelty And Significance:** 3
**Empirical Novelty And Significance:** 2
**Recommendation:** 5
**Confidence:** 4

**Main Review:**

The perspective of this paper is somewhat novel compared with existing training-based methods. The presented solution would be beneficial in detecting noisy labels. The authors present their algorithm clearly and the theoretical analyses are reasonable.
The whole method is based on the assumption that good representations can distinguish the noisy and clean labels. But the representations of hard samples are likely to be similar to the noisy ones. Although SimiRep achieves high F1-scores, the filtered hard samples can have a significant impact on model robustness. It might be necessary to conduct follow-up experiments on the model performance without filtered samples of SimiRep.
There are some mistakes in this paper. In the third line from the last of section 3.1, “the noisy labels are referred in representation learning”. It’s not true and should be “are not referred”. In the first line of page 5, the example of y_n3 is written into “0.0.33”.

**Summary Of The Paper:**

This paper proposed a training-free solution by using only good representations to detect noisy labels. The author designed two detection methods of voting and ranking to filter instances that are likely to be corrupted. The major contribution is proposing a training-free solution to efficiently detect noisy labels, which is different from most current methods. Also, theoretical analysis is conducted on the worst-case error bound and the choice of K.

**Summary Of The Review:**

The perspective is novel, and the presented solution would be beneficial in detecting noisy labels. But the significance of detecting noisy labels is uncertain and needs more experimental proof.

---

> ### Author Response · Authors · 2021-11-18
> **Response to Reviewer fRLd**
>
> Thanks for supporting the novelty of our paper and providing comments.
>
> We clarify the significance of data selection as follows. Basically, data quality is as important as (or even more important than) the algorithm. Detecting noisy labels helps to clean the dataset and improve data quality. The selected corrupted labels can be corrected by human workers. As mentioned in the third line of Introduction, **"Employing human workers to cleaning annotations is one reliable way to improve the label quality, but it is too expensive and time-consuming for a large-scale dataset."** To save human efforts, it is significant to design good algorithms to accurately select the corrupted instances.
>
> We also did experiments on Clothing1M to empirically demonstrate the effectiveness of our approach on large-scale datasets. Please see the response: **General Response to All Reviewers.**
>
> Other details are addressed as follows.
>
>
> **Question 1:** The representations of hard samples are likely to be similar to the noisy ones.
>
> **Response 1:**
>
> We did not define hard samples in our approach. To avoid confusion, let's define the samples with low confidence (or high entropy) k-NN labels as hard samples since the representation extractor cannot satisfy a good $(k,\delta)$ label clusterability. E.g., an instance with k-NN soft label $[0.5,0.5]$ is hard; an instance with k-NN soft label $[0.1,0.9]$ is easy. Note this is different from the definition of hard samples in the training-based methods. The hardness in their approach is defined wrt the training process: instances that are hard to be memorized. If hard samples are defined following the training-based works, their representations may be similar to noisy ones since noisy ones are usually hard samples in some synthetic noise models such as symmetric and asymmetric. But the so-called hardness of samples in our approach depends only on representation extractors and raw features, which should be independent of noisy labels. This nice property benefits from our training-free solution. Thus we cannot conclude the representations of hard samples in our approach are similar to the noisy ones.
>
> **Question 2:** Although SimiRep achieves high F1-scores, the filtered hard samples can have a significant impact on model robustness.
>
> **Response 2:**
>
> Yes, the filtered hard samples (defined in the previous response) are very important since the representation extractor has low confidence on them. We need to filter out and re-label them to ensure they are truly labeled. Note we cannot simply trust the model to re-label hard samples (that's also a potential problem in some existing label correction methods), therefore we do not explore these options in this paper, nor is this our focus. Following our detection, how would one correct the labels for the hard examples? We think this is a super interesting future question, and we believe a human-in-the-loop system might be helpful here.
>
>
> Recall the notion of hard samples in our context differs from noisy samples (due to the training-free property as explained in the above response): hard samples are determined by their representations -- those who do not satisfy the clusterability conditions are viewed as the hard examples. Intuitively, using our approach, the score of hard samples (if normalized to $[0,1]$) for the majority vote in Line 20 of Algorithm 1 would be around $0.5$ since it is unclear whether its k-NN share the same true label or not - scores close to 0.5 indicates high uncertainty in inferring its true label class.
>
> On the other hand, noisy labels are due to corruption on the labels. These two might correlate with each other but it's important to distinguish them.  Our method can distinguish between easy \& noisy samples and hard \& noisy samples.
>
> **Question 3:**
> It might be necessary to conduct follow-up experiments on the model performance without filtered samples of SimiRep.
>
> **Response 3:**
>
> Based on the above two responses, simply filtering out hard \& noisy samples may not be the best choice. But we also agree with Reviewer 1co7 that simply removing the corrupted samples and re-train the model may be an easy way to roughly check the performance. Please see the detailed results on Clothing1M in the response **General Response to All Reviewers**.
>
> **Others:** Thanks for showing typos. Fixed.

---

### Author Response · Authors · 2021-11-18
**General Response to All Reviewers**

We thank all the reviewers for their detailed and helpful comments. Several comments have motivated additional experiments to evaluate our method better. Following the suggestions from Reviewer 1co7, we tested our performance on Clothing1M. We also revised our submission based on the comments. The major changes are highlighted in blue. We will summarize our experiments on Clothing1M here. Other questions are replied to each reviewer individually. Please feel free to let us know if there is still any confusion.

**Experiments on Clothing1M:**

For Clothing1M (Xiao et al., 2015), we **could not calculate the F1-scores due to the lack of ground-truth labels - this was the primary reason that we didn't test on it in the original submission**. But we agree with the reviewers that performing an ablation study by removing the detected noisy labels can serve as a proxy for the detection performance.
We do the following experiment only with 1M noisy samples: 1) Implement our approach on Clothing1M with different representation extractors, and 2) train (with standard ResNet50 pre-trained on Imagenet as initialization) only on the selected clean dataset with **standard cross-entropy loss and standard data augmentations** (random crop, flip, normalization). We **did not apply any other sophisticated training techniques**, e.g., mixup (Zhang
et al., 2018), dual networks (Li et al., 2020b; Han et al., 2018), loss-correction (Liu \& Tao, 2015;
Natarajan et al., 2013; Patrini et al., 2017), and robust loss functions (Liu \& Guo, 2020; Cheng et al., 2021; Zhu et al., 2021a; Wei \& Liu, 2021).
The results are summarized as follows (Table 4 in our revised version).

**Table 4: Experiments on Clothing1M.** None: Standard training with 1M noisy data. R50-Img (or
ViT-B/32-CLIP, R50-Img Warmup-1): Apply our method with ResNet50 pre-trained on ImageNet
(or ViT-B/32 pre-trained by CLIP, R50-Img with 1-epoch warmup). Top-1 is **bold**.

|       Data Selection       | # Training Samples | Best Epoch |   Last 10 Epochs   | Last Epoch |
|:--------------------------:|:------------------:|:----------:|:------------------:|:----------:|
|            None            |      1M (100%)     |    70.32   | 69.44 $\pm$ 0.13 |    69.53   |
|      R50-Img     |    770k (77.0%)    |    72.37   | 71.95 $\pm$ 0.08 |    71.89   |
|        ViT-B/32-CLIP       |    700k (70.0%)    |    72.54   | 72.23 $\pm$ 0.17 |    72.11   |
| R50-Img Warmup-1 |    767k (76.7%)    |   **73.64**   | **73.28 $\pm$ 0.18** |    **73.41**   |

The selected indices will be released after acceptance.

---

### Decision · Program_Chairs · 2022-01-20

**Decision:**

Reject

**Comment:**

This paper proposed a training-free method to detect noisy labels based a given pretrained representation. The author suggested two methods based on either voting or ranking to filter noisy instances with corrupted labels without training. Reviewers generally agree that the technical novelty and contributions are only limited or marginally significant. Also experiments are not very convincing as there lacks of extensive comparisons with many existing methods for either noisy label removal or learning with noisy labels and the datasets are somewhat simple and not complex enough.